# Antiglycation Activity and HT-29 Cellular Uptake of Aloe-Emodin, Aloin, and *Aloe arborescens* Leaf Extracts

**DOI:** 10.3390/molecules24112128

**Published:** 2019-06-05

**Authors:** Guglielmina Froldi, Federica Baronchelli, Elisa Marin, Margherita Grison

**Affiliations:** Department of Pharmaceutical and Pharmacological Sciences, University of Padova, 35131 Padova, Italy; fe.baronchelli@gmail.com (F.B.); elisa.marin91@gmail.com (E.M.); margherita.grison@gmail.com (M.G.)

**Keywords:** cellular uptake, BSA assay, MTT assay, DPPH assay, antiglycation activity, plant extracts

## Abstract

*Aloe arborescens* is a relevant species largely used in traditional medicine of several countries. In particular, the decoction of leaves is prepared for various medicinal purposes including antidiabetic care. The aim of this research was the study of the antiglycation activity of two *A. arborescens* leaf extracts and isolated compounds: aloin and aloe-emodin. These phytoconstituents were quantitatively assessed in methanolic and hydroalcoholic extracts using high performance liquid chromatography (HPLC) analysis. In addition, the total phenolic and flavonoid contents were detected. In order to study their potential use in diabetic conditions, the antiglycation and antiradical properties of the two extracts and aloin and aloe-emodin were investigated by means of bovine serum albumin (BSA) and 1,1-diphenyl-2-picryl-hydrazil (DPPH) assays; further, their cytotoxicity in HT-29 human colon adenocarcinoma cells was evaluated by 3-[4,5-dimethylthiazol-2-yl]-2,5-diphenyl tetrazolium bromide (MTT) assay. Furthermore, the ability of aloin and aloe-emodin to permeate the cellular membranes of HT-29 cells was determined in order to estimate their potential in vivo absorption. This assessment indicated that aloe-emodin can substantially pass through cell membranes (~20%), whereas aloin did not permeate into HT-29 cells. Overall, the data show that both the methanolic and the hydroalcoholic *A. arborescens* extracts determine significant inhibition of glycation and free-radical persistence, without any cytotoxic activity. The data also show that the antiglycation and the antiradical activities of aloin and aloe-emodin are lower than those of the two extracts. In relation to the permeability study, only aloe-emodin is able to cross HT-29 cellular membranes, showing the attitude to pass through the intestinal layer. Overall, the present data surely support the traditional use of *A. arborescens* leaf extracts against hyperglycemic conditions, while aloin and aloe-emodin as potential drugs need further study.

## 1. Introduction

Since ancient times, plants have been primary sources of therapeutic preparations; among these, the genus *Aloe* is generating more and more interest in the scientific field [1,2]. *Aloe arborescens* Miller (Xanthorrhoeaceae), also known by the common name of “candelabra Aloe”, is a traditional medicinal plant native of central-southern Africa, and is very popular in South Africa, Asia, Russia, and Japan as an ornamental and medicinal plant [3]. In Italy, *Aloe* spp. are cultivated for decorative, cosmetic and medicinal purposes. *A. arborescens* plants are characterized by a central woody trunk that could reach a height of some meters at maturation. They have branches with bushy shrubs and long, thin, spiny, and grey-green leaves (Appendix A). The characteristic components of the red sap are 1,8-dihydroxyanthraquinones and their glycosides, mainly aloe-emodin and aloin, whose structures are reported in Figure 1. Further, a large amount of acemannan is present in the inner gel of the leaves [4]. The *Aloe* genus is known for its traditional use, but the isolated compounds, mainly aloe-emodin, have also been studied as potential conventional drugs. Aloe-emodin has been identified in several species of *Aloe* and in other genera such as *Rheum*, *Rumex*, and *Cassia* [5]. Aloin is the major anthraquinone of *Aloe*, which is a mixture of two diastereomers: aloin A (10S) and aloin B (10R) [6]. It is a C-glycoside that in vivo can be hydrolyzed in the gut to form aloe-emodin anthrone, which auto-oxidizes to aloe-emodin [7].

Diabetes mellitus (DM) is a metabolic disorder mainly characterized by chronic increase of blood glucose levels. To explain the association between hyperglycemia and the long-term complications of DM, one of the most important mechanisms proposed is the formation of advanced glycation end products (AGEs), which are triggered by endogenous processes related to hyperglycemia and oxidative stress [8]. Since it is known that glycation is accelerated by radical species, the influence of *A. arborescens* extracts, aloin and aloe-emodin on the bovine serum albumin (BSA) glycation and the free-radical persistence was investigated by means of BSA and 1,1-diphenyl-2-picryl-hydrazil (DPPH) assays.

Furthermore, it is interesting to note that *A. arborescens* has been proposed as a palliative therapy in patients with metastatic cancer even in co-administration with the conventional anticancer therapy [9]. In relation to this, the isolated compounds and in particular aloe-emodin have been evaluated in various types of human cancer cell lines where they showed antiproliferative activity [10,11,12]. This activity was also reported in vivo in various mouse models [13,14,15]. In general, different studies suggest the pharmacological potential of *Aloe* extracts and of its isolated compounds, but further studies are required for their actual medicinal use. With this idea, two types of extracts were prepared from the whole leaves of *A. arborescens*, as: (1) a methanolic extract, and (2) a hydroalcoholic extract obtained with ethanol 5% *v*/*v*, an alcohol percentage similar to that present in many “drinks” used in unconventional medicine. Thus, the two extracts were firstly assessed detecting their aloin and aloe-emodin amounts with high performance liquid chromatography (HPLC) analysis. Further, the total phenolic and flavonoid contents (TPC and TFC, respectively) were evaluated. Afterwards, the antiglycation and antiradical activities of aloin, aloe-emodin, and of the extracts were investigated and, in order to study the cellular cytotoxicity, their influence on HT-29 human colon adenocarcinoma cells were investigated using the 3-[4,5-dimethylthiazol-2-yl]-2,5 diphenyl tetrazolium bromide (MTT) assay. Lastly, to find answers to the questions: “Are anthraquinones adsorbed by the intestinal layer?” and “Could anthraquinones act in the cells?”, the quantities of aloin and aloe-emodin in HT-29 cells and in their culture medium were measured with a suitable experimental protocol.

Thus, the goals of this research were the identification and the quantification of aloe-emodin and aloin in the methanolic and the hydroalcoholic extracts of *A. arborescens* leaves, and the investigation of free-radical scavenging capacities and antiglycation properties of isolated compounds and *Aloe* leaves extracts. Lastly, the cytotoxicity and the capacity of aloin and aloe-emodin to cross the HT-29 cellular membranes were assessed.

## 2. Results

### 2.1. Methanolic and Hydroalcoholic Extracts: Phytochemical Characterization

Two extracts of *A. arborescens* were prepared from fresh leaves previously homogenized and lyophilized (yield: 4.8 ± 0.1% *p*/*p*). A weighed part of freeze-dry product was soaked in absolute methanol or 5% *v*/*v* ethanol for one hour, and then each solution was filtrated and stored at 4 °C, in the dark, and used within five days. The hydroethanolic preparation was chosen because similar to that used in folk medicine against cancer and proposed, even on the online market, without scientific evidence of effectiveness.

#### 2.1.1. Quantitative Detection of Aloin and Aloe-Emodin in the Extracts

To determine the aloin and aloe-emodin contents, both the extracts were analyzed with an HPLC instrument (Waters, Milan, Italy); exemplificative chromatograms are reported in Appendix A. The HPLC analysis showed that aloin is the component in the highest quantity in both extracts, as reported in Table 1. Moreover, the levels of aloin were the same in the two extracts, whereas the content of aloe-emodin was significantly lower in the hydroalcoholic extract. These results suggest that absolute methanol and 5% *v*/*v* ethanol have similar capacity to extract aloin from the lyophilized leaves, whereas aloe-emodin can be isolated from plant tissue to a higher quantity using methanol. The estimated aloin and aloe-emodin levels are respectively 0.1% *w*/*w* and 0.001% *w*/*w* of fresh *A. arborescens* leaves.

#### 2.1.2. Determination of Total Phenolic Content (TPC) and Total Flavonoid Content (TFC)

The TPC and TFC amounts were detected in both the methanolic and the hydroalcoholic extracts as reported in Table 2. The hydroalcoholic extract is characterized by a higher phenolic amount (16.8 µg gallic acid equivalents (GAE)/mg) compared to the methanolic extract (13.9 µg GAE/mg), while the flavonoid levels were similar in the two products. Thus, the 5% *v*/*v* ethanol/water solvent seems more suitable for phenolic extraction than methanol, while the use of the two different solvents did not change the amount of flavonoids extracted.

### 2.2. Potential Antidiabetic Activities of Methanolic and Hydroalcoholic Extracts, Aloin, and Aloe-Emodin

To study the potential antidiabetic property of *A. arborescens* extracts, the antiglycation activity was estimated using the BSA assay and, further, an antiradical assay was performed.

#### 2.2.1. Bovine Serum Albumin (BSA) Assay

BSA assay was performed to study the capacity to reduce the AGE formation of aloin, aloe-emodin, and the methanolic and hydroalcoholic extracts. Aloin and aloe-emodin were studied in the concentration range of 5–50 µM, showing a similar trend; both the anthraquinones significantly decreased the glycated albumin at the concentration of 25 µM (see Figure 2). Aloin and aloe-emodin at 50 µM reduced glycation by 12.8% and 19.5%, respectively. Likewise, the two extracts significantly inhibited the glycation; the effect became significant at the concentration of 500 µg/mL. The methanolic extract showed the highest activity, reducing glycation by 39.5% at the concentration of 1 mg/mL. Instead, the hydroalcoholic extract, at the same concentration, reduced glycated BSA only by 17.3%.

#### 2.2.2. DPPH Assay

The free-radical scavenging activity of aloin, aloe-emodin, and the two extracts was detected with DPPH assay based on the single electron transfer (SET) mechanism; it is an easy method, largely used to study antioxidant activity [16]. Ascorbic acid and quercetin were used as positive controls. As shown in Figure 3, aloin and aloe-emodin exhibited weak scavenging effects, 20% inhibitory concentration (IC_20_) >0.5 mM, whereas both the methanolic and the hydroalcoholic extracts showed significant DPPH radical scavenging effects; at the concentration of 1 mg/mL, quenching activities were respectively of 51.5 ± 3.2% and 64.0 ± 2.9%; the estimated half maximal inhibitory concentration (IC_50_) values were 1.0 mg/mL and 0.66 mg/mL (*p* < 0.05), respectively. These results suggest that the antiradical activities of the extracts are likely not due to the presence of the two anthraquinones because these ones showed only very slight activity.

### 2.3. Influence of Methanolic and Hydroalcoholic Extracts, Aloin, and Aloe-Emodin on HT-29 Cells Vitality

Since authors evidenced similar protein expression in the HT-29 cell line and human intestinal epithelium [17], this cell line was used to assess the influence of the extracts and single compounds on cellular vitality.

#### MTT Assay

The influence of 0.1–50 µM aloin and aloe-emodin and 0.1 µg/mL–1.0 mg/mL extracts on HT-29 cells viability after 24 h of incubation was studied. Paclitaxel (Sigma-Aldrich, Milan, Italy) was used as positive control. Aloe-emodin significantly reduces cellular viability from 10 µM while aloin was active only at 50 µM, as reported in Figure 4. The calculated IC_50_ value of aloe-emodin is 42.1 µM; thus, it manifested cytotoxic effects even though not comparable with the activity of paclitaxel, a well-known anticancer drug. On the other hand, neither of the two extracts showed cytotoxic effects even at the concentration of 1.0 mg/mL. It is possible to argue that neither the methanol nor the hydroalcoholic extract affect the viability of the HT-29 cell line.

### 2.4. Cellular Uptake of Aloin and Aloe-Emodin

To evaluate the potential bioavailability of aloin and Aloe-emodin, the capacity of the two anthraquinones to pass into the cells through the cellular membranes was detected.

#### Cellular Uptake in HT-29 Cells

The cellular permeability of aloin and aloe-emodin was evaluated using an original experimental protocol; each anthraquinone was added to the cellular medium at the concentration of 5 µM and incubated for 3 h in cultured HT-29 cells. The amounts of each compound were then assessed in the extra- and the intracellular compartments, as reported in Table 3. After incubation, the extracellular concentrations of aloin and aloe-emodin were of 4.12 and 0.15 µM, respectively, while the intracellular amount of aloe-emodin was of ~0.20 nmol/mg. In fact, aloin was not detected in the intracellular solution. The intracellular concentration of aloe-emodin was about 1/5 of the initial quantity. From these results, it is possible to conclude that this compound is able to pass, at least in part, the cellular membranes. On the other hand, the detection of aloin only in the extracellular environment suggests that this molecule cannot permeate cellular membranes. The differences between the initial concentrations and those measured in the intra- and the extracellular samples for both substances lead to the hypothesis that the two compounds were in part metabolized during the time of incubation. The HPLC-DAD analysis supported this observation because chromatograms confirm the presence of various peaks which may refer to unknown metabolites (Appendix A). Further, some biases for example due to compound sequestration from non-specific binding sites or other types of interactions may have occurred.

## 3. Discussion

*A. arborescens* is a medicinal plant of great interest in medicine both as herbal remedy and as source of isolated compounds such as aloin and aloe-emodin. On this subject, several studies are present in the literature but the consensus on its use is elusive and further studies are useful to define its actual role as medicinal plant. In this context, this study investigated two extracts obtained from the whole leaves of *A. arborescens* and their two characteristic components aloin and aloe-emodin. The total flavonoid and phenolic contents of extracts were detected and, with HPLC analysis, the identification and quantification of the two anthraquinones in each extract were performed. The antiradical and antiglycation properties of the extracts, aloin, and aloe-emodin were assessed; further, their influence on cell vitality was studied. Finally, the capacity of aloin and aloe-emodin to cross cellular membranes was evaluated in HT-29 cells, used as model of intestinal layer.

In general, among phenols, aloin and aloe-emodin are distinctive compounds of *Aloe* extracts often used as phytochemical markers; concerning this genus, the average content of aloin is high in *A. arborescens* [18]. In both extracts, the aloin quantity detected was about 20 mg/g in lyophilized leaves, and 0.1% *p*/*p* in fresh leaves. Moreover, the aloe-emodin content was always considerably lower than the aloin level, and also was significantly lower in the hydroalcoholic extract compared to the methanolic one, as reported in Table 1. In the literature few studies are published on this topic; recently, authors reported that the amount of aloin was of 2.9% *p*/*p* in *A. arborescens* leaf exudate (which is a more concentrated preparation) [18]. Overall, other studies reported very low or even no amounts of aloe-emodin in extracts of different *Aloe* species [19], as well as of *A. arborescens* [18].

Aloin is the C-glucoside of aloe-emodin anthrone and it is present in nature as two diastereomers, aloin A and aloin B, as reported in the HPLC chromatogram in Appendix A. It is of interest to note that aloin is a marker used to identify *Aloe* spp. [20]. Even if aloin is known as a laxative agent [21], nowadays new pharmacological properties are being investigated. A recent study supports the anticancer activity of aloin, showing that it was able to induce apoptosis of A549 cells [22]. Further, other authors demonstrated that the treatment of bone marrow-derived mesenchymal stem cells with aloin affects bone metabolism and promotes osteogenic differentiation, enhancing mineralization and alkaline phosphatase activity and involving the ERK1/2-Runx2 signaling pathway [23]. Regarding aloe-emodin, in vitro studies have evidenced various properties like antiplatelet [24] and anticancer activities [25].

Studies on the potential activity of plant extracts or isolated compounds against AGE formation could discover new agents with antiglycation activity useful in the prevention of hyperglycemic damage. Since glycated albumin is associated with diabetic complications [26], the BSA assay is a valuable test to assess the potential antidiabetic property of products. The present results showed significant antiglycation activities of both methanolic and hydroalcoholic *A. arborescens* extracts at the concentration of 500 µg/mL. The isolated compounds aloin and aloe-emodin reduced albumin glycation from the concentration of 25 µM. Very few data are available on effects induced by *Aloe* and its constituents on protein glycation; aloe-emodin and *Aloe sinkatana* extracts decreased protein–sugar reactions, even if aloe-emodin exhibited only weak activity in BSA-glucose assay, with an IC_50_ of 517 µM [27]. Other authors suggested that *Aloe vera* and aloin could protect the enzyme superoxide dismutase against the structural changes induced by glycation [28]. Previously, a protective role of *A. arborescens* was observed on islets of Langerhans in mice and an inhibitory effect was measured on the jejunal glucose absorption [29]. Even if the scientific literature reports several articles on *Aloe* species and antidiabetic activity, no studies have reported on *A. arborescens* and protein glycation damage. Thus, the present data show for the first time the antiglycation activity of *A. arborescens* leaf extracts.

Due to the strong relationship between glycation and oxidative stress, one being the cause of the other [30], the antioxidant properties of the two extracts and aloin and aloe-emodin were evaluated with DPPH assay. Both extracts evidenced a concentration-dependent antiradical activity, which was significant from 50 µg/mL, while the single anthraquinones showed very low activity. Recently, studies reported a fair antioxidant activity of *A. arborescens* extracts compared with other *Aloe* species [18], in particular against the oxygen radicals [31]. In agreement with the present results, the relatively small contribution of aloin and in general of anthraquinones to the overall radical scavenging activity has been reported also by other researchers [32,33,34]. Several authors ascribe the antiradical activity of plant extracts to the presence of phenols; thus, we detected the phenols and flavonoids amounts in the extracts. The hydroalcoholic extract had a higher phenolic amount (16.8 µg GAE/mg) than the methanolic extract (13.9 µg GAE/mg), suggesting that the 5% *v*/*v* ethanol/water solvent can be considered a suitable solvent for the extraction of phenols from *Aloe* leaves. Overall, phenolic compounds are generally linked with health benefits, for example in cardiovascular diseases and cancer, and their presence is deemed helpful for disease prevention [35,36].

Moving from in vitro to in vivo, one of the critical points is the bioavailability of the studied compounds. The HT-29 cell line, obtained from human colorectal adenocarcinoma, is a suitable model for bioavailability measurements thanks to its similarities, both in phenotype and enzymatic expression, with mature intestinal cells like enterocytes [17]. In the present investigation, firstly the potential cytotoxicity was evaluated. The two extracts did not show any cytotoxic activity up to 1 mg/mL, while aloin and aloe-emodin significantly reduced cell viability respectively from 50 µM and 10 µM. In details, aloe-emodin was the most active, causing a concentration-dependent inhibition in the range of 10–50 µM; at 10 µM aloe-emodin reduced vitality to 75.6%. In agreement with these findings, authors [37] showed that aloe-emodin inhibited HT-29 viability in a concentration- and time-dependent manner, reducing it to 82.8% after 24 h of treatment at 10 µM. Other authors showed anticancer activities of aloe-emodin in different cell lines, like U87MG from human glioblastoma and T24 from human bladder cancer [38,39]. Some studies showed that also aloin exerts anticancer properties, inhibiting both cell proliferation and angiogenesis. In detail, aloin was able to decrease the vascular endothelial growth factor (VEGF)-induced viability of human umbilical vein endothelial cells (HUVECs) from the concentration of 10 µM which was not cytotoxic in the same cell line, under normal conditions. A direct measurement in colorectal cancer cell lines showed an inhibitory effect on cell viability with an IC_50_ of 200–240 µmol/L [40].

Although HT-29 cells are a useful model of human enterocytes, these may behave differently from human intestinal mucosa cells [17]. However, an estimation of oral bioavailability of aloe-emodin and aloin can be obtain via cellular uptake assay using HT-29 cells [17]. Aloin and aloe-emodin 5 µM, incubated for 3 h in HT-29 cells, were assessed both in the extra- and the intracellular compartments. Aloin was almost totally found in the extracellular solution (4.12 μM) whereas it was not detected in the intracellular compartment. On the contrary, aloe-emodin was detected in a very low amount in the extracellular solution (0.15 μM), whereas a significantly higher amount was found at the intracellular level: 0.20 nmol/mg protein—about 1 μM. These results show that only aloe-emodin can pass through cellular membranes into the cells. Thus, it could be concluded that aloe-emodin has higher bioavailability than its C-glycosyl compound aloin. However, further in vitro assessments, e.g., using intestinal epithelial cell monolayers, and also in vivo studies are necessary to evaluate aloe-emodin and aloin bioavailability. Previously, other authors studied the cellular uptake of aloe-emodin into a human breast adenocarcinoma cell line (MCF-7) at 5 µM with a 24 h incubation, showing an intracellular uptake of 1.4 μg/mg protein [41]. Moreover, in agreement with present results, other authors compared the aloin and aloe-emodin uptake in the Caco-2 cell monolayer and everted gut sac model, showing that the absorption rate of aloe-emodin is always higher than those of aloin [42]. Different mechanisms of transport are suggested for the two molecules. Since aloe-emodin is an aglycone, it could permeate membranes through passive diffusion, while aloin uptake requires glucose transporters like sodium-dependent glucose co-transporter (SGLT1) and GLUT2 glucose transporter [43]. Not only transportation but also metabolism may affect the bioavailability of the anthraquinones in vivo. Altogether, the literature and the present experimental data suggest that aloin may be metabolized into aloe-emodin in the intestine and then absorbed into the bloodstream, giving also systemic effects. Pharmacokinetic studies carried out in rats showed that intraperitoneal administration of aloe-emodin at 7.5 mg/kg daily for six days determined higher biodistribution into tissues with rich blood supply such as the spleen, lung, heart, and liver [41]. Recently, authors showed in high-fat-fed rats treated with 100 mg/kg aloe-emodin intragastrically administered each day for six weeks, significant antiarrhythmic effects, preventing QT prolongation [44].

## 4. Materials and Methods

### 4.1. Extract Preparation

Mature leaves from 42 month-old plants were collected in the month of June 2016 from a northern Italy cultivation (Manerba del Garda, Brescia). A dried sample of leaves is deposited at the Department of Pharmaceutical and Pharmacological Sciences, Laboratory of Pharmacognosy, with the voucher number of AA-BS-011. The leaves were cut to the base and cleaned with a wet cloth, while the side thorns were removed. Subsequently, leaves were subdivided into small pieces to be homogenized in a mixer, obtaining 463 g of product which was then lyophilized up to complete dehydration with a freeze dryer (E2M18 Edwards, UK). The freeze-dried product was stored at −20 °C. To obtain the two extracts, a weighed part of product was soaked in methanol or in 5% *v*/*v* ethanol for 1 h; then, each solution was filtered and stored at 4 °C in the dark, and used within five days. The stability of solutions during the days was tested with TPC and TFC assays and the quantitative detection of aloin and aloe-emodin contents with HPLC analysis.

### 4.2. High Performance Liquid Chromatography (HPLC-DAD) Analysis

Chromatographic analyses were carried out by the HPLC instrument (Waters Corporation, Milan, Italy) equipped with 1525 Binary HPLC pump (Waters Corporation, Milan, Italy) and 2998 Photodiode Array Detector (Waters Corporation, Milan, Italy). The chromatographic separation was performed with a Symmetry® RP C18, 4.6 × 75 mm, 3.5 μm column (Waters Corporation, Milan, Italy). The method was modified from that of [20]; a linear methanolic gradient from 20 to 80% *v*/*v* in 15 min (A: water with acetic acid 0.5% *v*/*v*, and B: methanol with acetic acid 0.5% *v*/*v*), with a flow rate of 1 mL/min was applied. Peaks were detected in the range of 210–400 nm. All samples were filtered by membrane filters (0.22-μm pore size, Millipore, Burlington, MA, USA) and then injected (10 μL). The compound identification was performed on the basis of retention time and spectral matching with aloin and aloe-emodin standards. To improve the qualitative and quantitative identification of aloin and aloe-emodin in the two extracts, in a set of analyses a precise amount of each standard was co-injected with each extract for the HPLC detection. These ones confirmed the retention time of 9.5 and 9.8 min for aloin B and A, respectively, and of 12.3 min for aloe-emodin, as reported in Appendix A. For the preparation of the calibration curves, standard stock solutions were prepared in methanol, filtered and appropriately diluted to obtain the desired concentrations (1.0, 5.0, 10.0, and 20.0 μM). The calibration graphs were plotted as the linear regression of the peak areas versus concentrations. To determine the aloin and aloe-emodin contents, both extracts were analyzed with HPLC-DAD instrument by injection of each extract, at the concentration of 1 mg/mL. The amount of each compound in extra- and intracellular solutions was calculated comparing the results with the standard calibration curves. The last step included the analysis of the lysate protein content using the Lowry protein assay [45], to obtain the final results expressed as nmol of aloin or aloe-emodin per mg of protein.

### 4.3. Determination of Total Phenolic Content (TPC) and Total Flavonoid Content (TFC)

TPC was determined using the Folin-Ciocalteu reagent according to the method previously described [46]. TPC of each extract is expressed as µg of gallic acid equivalents (GAE) per g of freeze-dried product or as mg of gallic acid equivalents (GAE) per 100 g of fresh-weight (FW) product. TFC was detected by aluminum chloride (AlCl_3_) colorimetric method [47] and expressed as µg of quercetin equivalents (QE) per g of freeze-dried product or as mg of quercetin equivalents (QE) per 100 g of fresh-weight (FW) product.

### 4.4. DPPH Assay

The free radical-scavenging activity of samples was measured by 1,1-diphenyl-2-picryl-hydrazil (DPPH) method [48]. Samples solutions were prepared and added to 70 μM DPPH; the mixtures were kept in the dark for 60 min, and the absorbance was read at 517 nm using a Beckman Coulter DU 8005 spectrophotometer (Fullerton, CA, USA). The antiradical curves were obtained with different sample concentrations. Radical scavenging capacity is expressed as percentage effect (E %).

### 4.5. Bovine Serum Albumin (BSA) Assay

The preparation of glycated BSA was performed according to a previously described method, with minor modifications [49]. Shortly, AGEs, which are the product of glycation, were determined using BSA, as protein substrate, and glyoxal, as glycating agent. The solutions of substrates were added with each extract (50, 100, 500, and 1000 µg/mL), or aloe-emodin or aloin (5, 10, 25 and 50 µM), and incubated at 37 °C for 7 days. After that time, the fluorescence measurement allows the calculation of glycation inhibition as per cent difference between control condition and glycation in the presence of the samples. Aminoguanidine (50 mM, AG) was used as positive control.

### 4.6. 3-[4,5-Dimethylthiazol-2-yl]-2,5 Diphenyl Tetrazolium Bromide (MTT) Assay

The cellular viability after the treatment with the selected compounds or the extracts was assessed with the MTT assay [50]. HT-29 cells were seeded in a 96-well plate at the density of 5000 cells in each well and let grow overnight; the day after cells were treated with different concentrations of each substance and after 24 h an aliquot of MTT solution was added to each well, reaching the final concentration of 500 μg/mL. After the reduction of the MTT by cellular enzymes, the medium was removed and the insoluble formazan salts were solubilized with an acid solution of 2-propanol. The absorbance of each purple formazan solution was measured using a Victor multilabel plate reader (PerkinElmer, MA, USA), setting the wavelength at 520 nm.

### 4.7. Cellular HT-29 Uptake Assay

To assess the aloin and aloe-emodin cellular uptake, the human colon adenocarcinoma cells (HT-29) were seeded in 6-well plates in complete medium and let grow until confluence for 48 h. Then, culture medium was removed and cells were washed with PBS and treated for 3 h with 5 µM aloin or aloe-emodin. After the incubation, an aliquot of the extracellular solution was taken from each well and stored at −20 °C for further analysis. Then the medium was removed, cells were washed two times with PBS and prepared for the following steps: (1) protein quantification, where cells were lysed with 200 μL of lysis buffer and wells were washed with 200 μL of PBS; and (2) cellular uptake evaluation, where cells were collected gently scraping with PBS, centrifuged at 1250 rpm for 5 min, and added to 500 μL of 80% *v*/*v* methanolic solution in water/0.1% *v*/*v* acetic acid to extract the intracellular content. The samples were maintained in ice for 15 min, then sonicated and centrifuged at 10,000 rpm for 10 min. The precipitated proteins were excluded, while the supernatant solutions were collected for chromatographic analysis. The lysates and the intracellular solutions were stored at −20 °C for the HPLC-DAD analysis.

### 4.8. Statistical Analysis

Data are expressed as mean ± SEM of at least three experiments. The experimental data were graphed and analyzed using GraphPad Prism 5 (San Diego, CA, USA), and the half maximal inhibitory concentration (IC_50_) values were calculated [51]. The differences between control and treatment were assessed by Student’s t test. The level of significance was set at *p* < 0.05.

## 5. Conclusions

This research shows that both the methanolic and the hydroalcoholic *A. arborescens* leaf extracts have antiglycation and antiradical activities which could be ascribable to their phenolic components, and only in part to aloin and aloe-emodin. These two anthraquinones show moderate inhibitory activities on cell viability, whereas the two extracts completely lack cytotoxicity. Further, the capability of aloe-emodin to be absorbed into the cells was evidenced. Overall, the data from this research support the helpfulness of using *A. arborescens* leaf extracts against hyperglycemia-associated diseases.

## Figures and Tables

**Figure 1 molecules-24-02128-f001:**
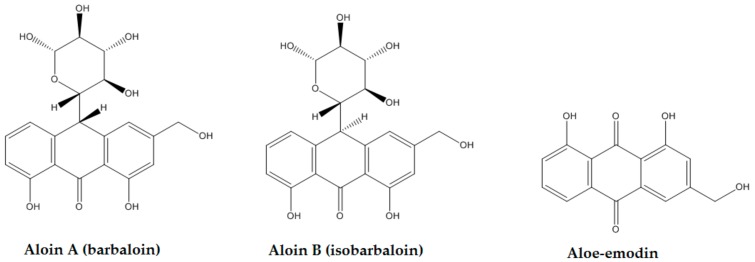
Aloin A, aloin B, and aloe-emodin structures.

**Figure 2 molecules-24-02128-f002:**
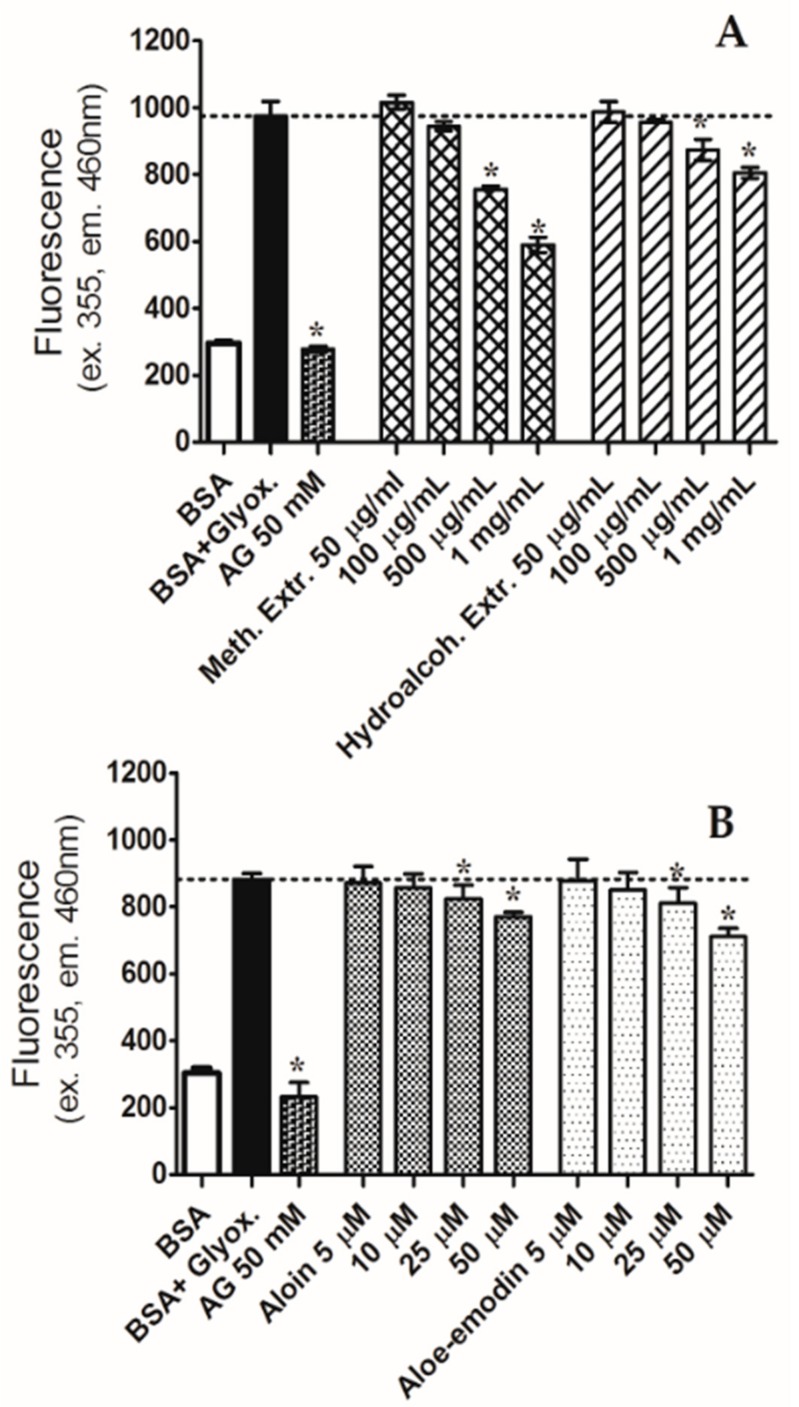
Effects of the methanolic (Meth.) and hydroalcoholic (Hydroalcoh.) *A. arborescens* extracts (**A**), and aloin and aloe-emodin (**B**) against bovine serum albumin (BSA) glycation after seven days of incubation with glyoxal (Glyox.) 0.4 mg/mL. Aminoguanidine (AG) was used as positive control. Extr: extract. * *p* < 0.05 vs. BSA plus glyoxal.

**Figure 3 molecules-24-02128-f003:**
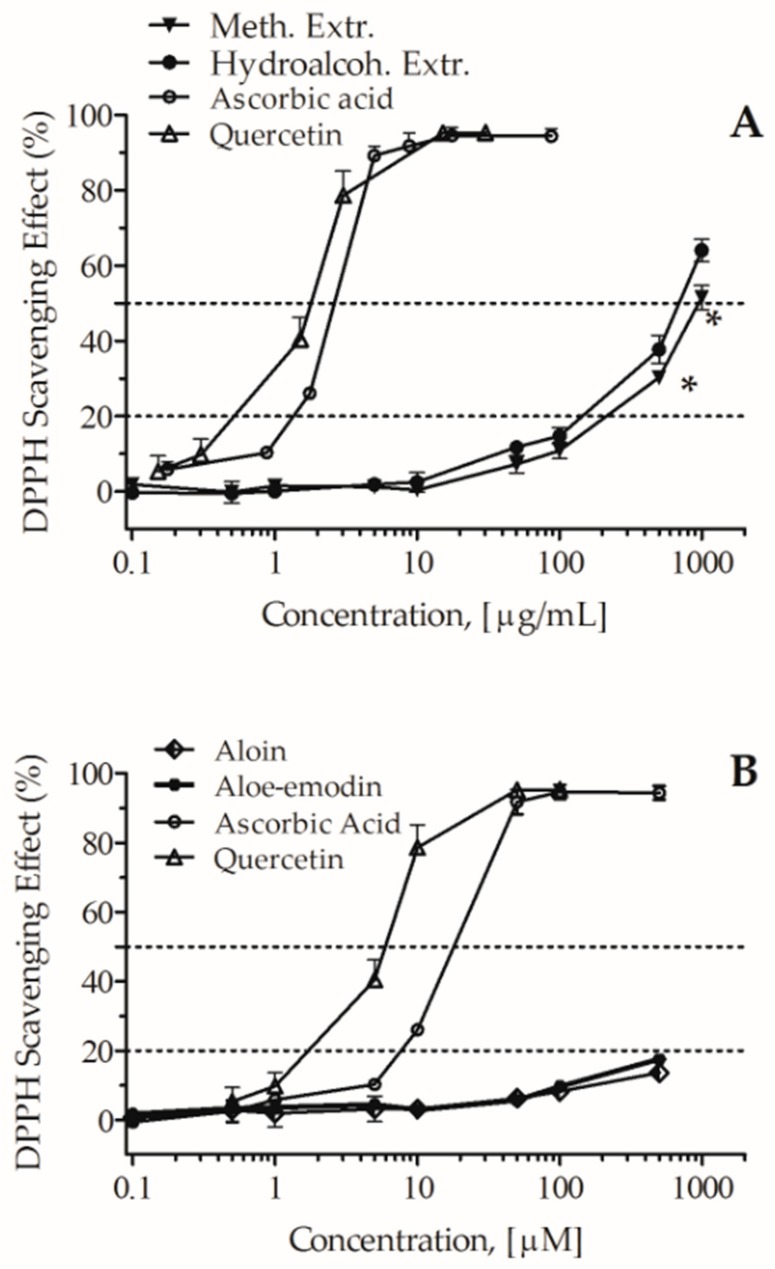
Antiradical activity of the methanolic (Meth.) and hydroalcoholic (Hydroalcoh.) *A. arborescens* extracts (**A**), and aloin and aloe-emodin (**B**). Ascorbic acid and quercetin were used as positive controls. Extr: extract. * *p* < 0.05 vs. hydroalcoholic extract.

**Figure 4 molecules-24-02128-f004:**
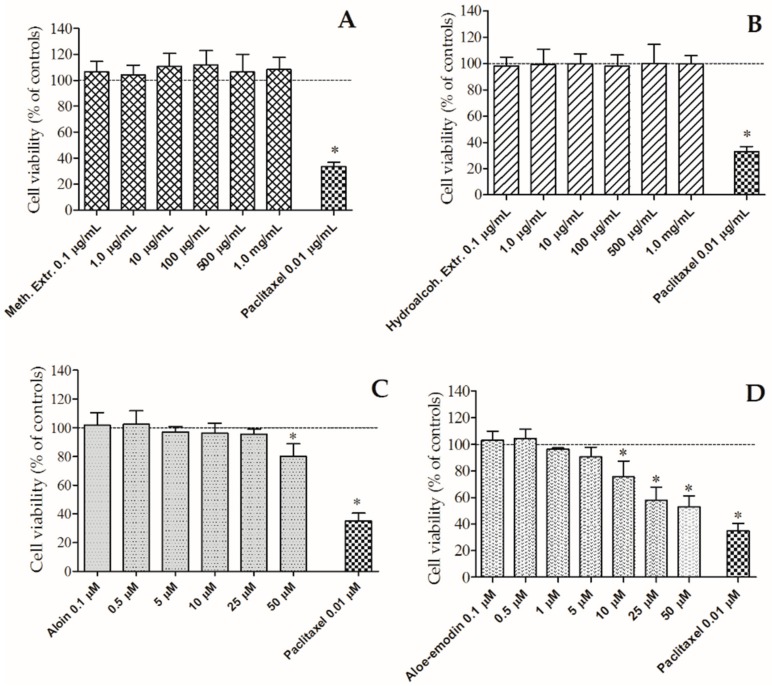
Activity of the methanolic (Meth.) (**A**) and hydroalcoholic (Hydroalcoh.) (**B**) extracts, and aloin (**C**) and aloe-emodin (**D**) on cellular viability of human colon adenocarcinoma (HT-29) cells. Paclitaxel was used as positive control. Extr: extract. * *p* < 0.05 vs. controls (cell viability without treatment).

**Table 1 molecules-24-02128-t001:** Aloin and aloe-emodin contents in *A. arborescens* leaf extracts.

*Aloe arborescens* Leaves	Aloin µg/mg ^§^	Aloe-emodin µg/mg ^§^	Aloin mg/100 g FW ^¤^	Aloe-emodin mg/100 g FW ^¤^
Methanolic extract	20.0 ± 1.2	0.22 * ± 0.1	96.7 ± 5.6	1.07 ± 0.05
Hydroalcoholic extract	21.9 ± 1.7	0.08 ± 0.1	106.1 ± 8.1	0.39 ± 0.01

^§^: dry extract; ^¤^: fresh weight of leaf homogenate; * *p* < 0.05 vs. aloe-emodin in the hydroalcoholic extract.

**Table 2 molecules-24-02128-t002:** Total phenolic and flavonoid contents of *Aloe arborescens* leaf extracts.

*Aloe arborescens* Leaves	TPC		TFC	
	GAE μg/mg	GAE mg/100 g FW	QE μg/mg	QE mg/100 g FW
Methanolic extract	13.85 ± 0.46 *	67.11 ± 2.31	3.42 ± 0.14	16.53 ± 0.68
Hydroalcoholic extract	16.84 ± 0.77	81.43 ± 3.71	3.09 ± 0.17	14.93 ± 0.83

TPC: total phenolic content; TFC: total flavonoid content; GAE: gallic acid equivalents; QE: quercetin equivalents; FW: fresh weight of leaf homogenate. * *p* < 0.05 vs. GAE in the hydroalcoholic extract.

**Table 3 molecules-24-02128-t003:** The extra- and the intracellular amounts of aloin and aloe-emodin in HT-29 cells in culture.

*Aloe arborescens* Anthraquinones	Concentration Inoculated	Intracellular Concentration	Extracellular Concentration
Aloin	5 μM	*	4.12 μM
Aloe-emodin	5 μM	0.20 nmol/mg (~1.0 μM)	0.15 μM

Each anthraquinone was incubated in human colon adenocarcinoma (HT-29) cells, for 3 h at 37 °C; then extra-and intracellular amounts were detected with high performance liquid chromatography analysis. * not detectable.

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
