# Peer review of "Antiglycation Activity and HT-29 Cellular Uptake of Aloe-Emodin, Aloin, and Aloe arborescens Leaf Extracts"

_molecules, 2019, doi:10.3390/molecules24112128_

Round 1
Reviewer 1 Report
Review MOLECULES
The manuscript investigated the anti-glycation of Aloe arborescens leaf extract and two main isolated compounds of theses extracts: Aloe-emodin and Aloin. Phenolic, flavonoid and both aloin contents were firtly determined in this extract. In addition, Antiglycative and antiradical activities in in vitro model of glycated BSA were evaluated. Finally, authors investigated the potential cytotoxicity and cellular uptake in adenocarcinoma HT-29 cells.
Even if this study is well conducted and shows interesting findings, this paper shows need to be improved and authors should take into account the following suggestion and remarks to improve the manuscript; Additional experiments and robust methods are needed to better characterize this phenomenon observed in this paper.
Results part:
1. Methanolic and Hydroalcoholic Extracts: Phytochemical Characterization : Authors mentioned that both leaf extract were stored at 4°C for 5 days. Could author authors specify whether the stability of the compounds is preserved and no variation no variation in terms of polyphenolic and flavonoid content is observed during this period of storage?
2.Quantitative detection of aloin and aloe-emodin in the extracts. There is estimation of aloin and aloe-emodin respectively 0.1% and 0.001% w/w of fresh leaves. That means that both molecules seem to be minor compounds of the extract. Could authors specify the proportion of these two compounds in the total polyphenolic content?
Regarding to the weak polyphenolic contribution of these molecules, it would be necessary to compare their antioxidant, antiradical activities with and other polyphenolic compounds of the extract. But as a first step this study should be completed by a more detailed description of the composition of the extracts
3. Bovine Serum Albumin (BSA) assay
Authors investigated antiglycative activity of aloin molecules and leaf extract by using AGE induced BSA models and assessing the reduction of glycation by fluorescence AGE determination. Firstly, in figure 3, authors should add the fluorescence of BSA alone without any glycative agent. Secondly, the determination of AGE level by fluorescence spectroscopy is not really a quantitative assay and this method gives only the relative level of fluorescent AGE. Alternative quantitative assay is needed to strengthen these results (AGE level by ELISA, Western blot or other biochemical assay…)
4. MTT assay
Authors evaluate the influence aloin and aloe-emodin on HT-29 cells viability after 24 h of incubation with these molecules. Cytoxicity of both molecules are observed for the highest concentration. Considering that the principle of MTT assay is based on the mitochondrial succinate dehydrogenase activity and the fact that some polyphenol molecules could affect mitochondrial metabolism (Int J Mol Sci. 2018 Sep 13;19(9). pii: E2757. doi: 10.3390/ Curr Med Chem. 2017 May 28. doi: 10.2174/), it is necessary to validate these results with another method (LDH assay, cell counting, apoptosis/necrosis or cell death detection assay).
5.DPPH assay. Through DDPH assay authors show a weak antiradical effect for low concentration Aloin molecules. It would be interesting to confirm this weak antioxidant activity with an alternative method using the in cellulo model of HT-29 cells with oxidative stress probe (DCFH-DA, Amplex Red or Mitosox…). This additional experiment will strengthen this work.
6.Cellular uptake in HT-29 cells. It is very interesting to evaluate the amount Aloin molecules uptaked by HT-29cells but why authors didn’t perform the experiment with the same experimental design as they did for MTT assay (24h of incubation) ? In this case, intracellular concentration of Aloin could be detected after a longer period of incubation.
In addition, it would be interesting to have a dose effect on this uptake with an increasing concentration of Aloin or Aloin-emodin.
7.Considering this last interesting result,the effect of plant extract and aloin molecules should on HT-29 cells metabolism should be in more detail investigated
Author Response
All suggestions have been seriously taken into account and, as far as feasible, the changes proposed were done. The manuscript changes are evidenced in the text and, each point, discussed below. Thanks for your valuable review which helped the manuscript revision.
Methanolic and Hydroalcoholic Extracts. Phytochemical Characterization: Authors mentioned that both leaf extract were stored at 4°C for 5 days. Could author authors specify whether the stability of the compounds is preserved and no variation in terms of polyphenolic and flavonoid content is observed during this period of storage?
We tested the stability of the extract solutions and the isolated compounds daily by the use of HPLC analysis; by this method, we verified that aloin and aloe-emodin peaks of chromatograms remain unchanged at least for 8 days in the considered solutions. We therefore decided to keep the solutions for 5 days: from Monday to Friday. This procedure has also been chosen based on the fact that various commercial preparations containing Aloe are stored for many days before the oral administration in patients. Furthermore, also the stability of the phenols and flavonoids was always checked throughout the 5 days. All these procedures showed that the solutions can be used for at least one week. We added the specifications requested in the manuscript (methods, P.10 R. 565-566).
Quantitative detection of aloin and aloe-emodin in the extracts. There is estimation of aloin and aloe-emodin respectively 0.1% and 0.001% w/w of fresh leaves. That means that both molecules seem to be minor compounds of the extract. Could authors specify the proportion of these two compounds in the total polyphenolic content? Regarding to the weak polyphenolic contribution of these molecules, it would be necessary to compare their antioxidant, antiradical activities with and other polyphenolic compounds of the extract. But as a first step this study should be completed by a more detailed description of the composition of the extracts.
Several researches have reported that Aloe leaves contain many types of secondary metabolites, such as phenolic acids, polyketides, chromones, anthrones, flavonoids and, in the inner leaf pulp, very high acemannan polysaccharide amount, P. 1-2 L. 42-54 (El Sayed et al., 2016; Salehi et al., 2018). Among all these, aloin is generally considered the most characteristic secondary metabolite of Aloe plants (Gutterman and Chauser-Volfson, 2000). For this reason, it is often used as the marker of this genus. Indeed, aloin is a C-glycosyl phenol chemically stable, with chemotaxonomic significance for Aloe species. In the present research, aloin and aloe-emodin were quantitatively detected by use of the HPLC analysis, whereas the amount of phenols was determined using the Folin-Ciocalteu reagent (Ainsworth and Gillespie, 2007). The Folin-Ciocalteu assay relies on the transfer of electrons in alkaline medium from phenolic compounds to phosphomolybdic/phosphotungstic acid complexes, then determined spectroscopically at 765 nm. The TPC is a quantification of phenols expressed as equivalents of gallic acid, which cannot be considered an absolute quantification, by the way hard to obtain because phenols have very various chemical structures.
Being the HPLC analysis and TPC quantification very different procedures, we did not relate directly the amounts of aloin and aloe-emodin to the total phenols content, even if these data can be obtained using few equations applied to values of tables 1 and 2. Using those, the amount of aloin was 50.7 - 58.6% of TPC, and that of aloe-emodin 0.29 - 0.99% of TPC.
“That is: 20.0 µg/mg aloin in the methanolic extract (Table 1) and 13,85 GAE µg/mg phenols (Table 2) → 20,0/418,398 (MW aloin) = 0.0478 µmol aloin → 13,85/170,12 (MW gallic acid) = 0.0814 µmol gallic acid (TPC) → 0,0478 x 100/0.0814= 58,65% - etc.”
For the above reasons, we did not report the percentage of aloin and aloe-emodin on TPC values in the manuscript.
Bovine Serum Albumin (BSA) assay. Authors investigated antiglycative activity of aloin molecules and leaf extract by using AGE induced BSA models and assessing the reduction of glycation by fluorescence AGE determination. Firstly, in figure 3, authors should add the fluorescence of BSA alone without any glycative agent. Secondly, the determination of AGE level by fluorescence spectroscopy is not really a quantitative assay and this method gives only the relative level of fluorescent AGE. Alternative quantitative assay is needed to strengthen these results (AGE level by ELISA, Western blot or other biochemical assay…)
Thank you very much for your suggestions. The histograms of BSA fluorescence alone were added in Figure 2 (past Figure 3), as requested.
About the other point, BSA assay is a very accepted assay, useful to study the glycation reaction because it mimics the physiological process of in vivo glycation. Furthermore, various authors suggested the glycated albumin level as a diagnostic marker useful in diabetic patients. Anyway, we agree with the reviewer that there are other assays to study proteins glycation, which however require more time and are relatively expensive; also on this view, we used the BSA assay in this study of Aloe arborescens extracts to obtain a first detection of the antiglycation activity of the substances studied.
MTT assay. Authors evaluate the influence aloin and aloe-emodin on HT-29 cells viability after 24 h of incubation with these molecules. Cytoxicity of both molecules are observed for the highest concentration. Considering that the principle of MTT assay is based on the mitochondrial succinate dehydrogenase activity and the fact that some polyphenol molecules could affect mitochondrial metabolism (Int J Mol Sci. 2018 Sep 13;19(9). pii: E2757. doi: 10.3390/ Curr Med Chem. 2017 May 28. doi: 10.2174/), it is necessary to validate these results with another method (LDH assay, cell counting, apoptosis/necrosis or cell death detection assay).
- The MTT assay is the most used assessment for cytotoxicity, cell viability, and proliferation studies in cell biology. It gives a yellowish aqueous solution which, on reduction by dehydrogenases present in metabolically active cells, yields a water insoluble violet-blue formazan. It is widely thought that the amount of MTT formazan is directly proportional to the number of living cells (Stockert et al., 2012). It has been claimed that the mitochondrial succinate dehydrogenase of viable cells reduced MTT to the corresponding formazan, even if mitochondria do not show reducing properties, but rather oxidizing power. Biochemical evidence indicates that MTT is mainly reduced in the cytoplasm by NADH (and NADPH) and dehydrogenases associated to the endoplasmic reticulum (Berridge et al., 2005).
Overall, independently of mechanisms bind to formazan production, we always carried out controls with all reagents without cells and also the standard blank (cells + reagents – substance) excluding all non-specific interference. Furthermore, the MTT assay is the reference method in the studies of cellular vitality of whatever compound. Besides the Aloe extracts, aloin and also aloe-emodin showed very low inhibition of cellular vitality, also our data agree with those of (Cheng and Dong, 2018), as reported in the discussion (P. 9 L. 514-519). For all these reasons, at the present, new experiments were not performed but we will take in mind the reviewer's suggestions when we will plan future studies. Thank you for the indications.
DPPH assay. Through DDPH assay authors show a weak antiradical effect for low concentration Aloin molecules. It would be interesting to confirm this weak antioxidant activity with an alternative method using the in cellulo model of HT-29 cells with oxidative stress probe (DCFH-DA, Amplex Red or Mitosox…). This additional experiment will strengthen this work.
We agree with the reviewer that it could be of interest to perform other experiments using other approaches to study antioxidant activity of aloin, aloe-emodin and the extracts, but in this first step of our study, we have chosen the DPPH assay that is suitable to test anti-radical activity of either isolated compounds and extracts.
By this test we observed only moderate activities; also for this reason we did not perform other set of experiments on antiradical activity. Thank you for the suggestion.
Cellular uptake in HT-29 cells. It is very interesting to evaluate the amount Aloin molecules uptaked by HT-29cells but why authors didn’t perform the experiment with the same experimental design as they did for MTT assay (24h of incubation) ? In this case, intracellular concentration of Aloin could be detected after a longer period of incubation.
Cellular uptake is a quick pathway by which substrates can cross cellular membranes; in this type of study, usually the incubation time is between 30 min and 3-4 h. In fact, longer time of incubation may give also the time to the studied compound to go back and further to be metabolised; for these reasons, we chose a short time of incubation. On the other hand, in the vitality studies of substances that are not highly cytotoxic the incubation time is always at least 24 h, sometimes 48 and 72 h too. This because is necessary a long time to modify cells proliferation, an event which requires hours. In brief, we chose different experimental conditions in the MTT and the cellular uptake assays because the aims were different, having these dissimilar biological characteristics.
In addition, it would be interesting to have a dose effect on this uptake with an increasing concentration of Aloin or Aloin-emodin.
Thank you for the suggestion; however, in this first step of our research on aloin and aloe-emodin, the main aim was to evaluate if the studied compounds were able or not to pass through the cellular membranes in vitro, in standard conditions; deeper pharmacokinetics studies could be carried out in the future taking in mind that they are very time-consuming. Furthermore, other in vivo studies could be appropriate evaluating the compounds bioavailability.
Considering this last interesting result, the effect of plant extract and aloin molecules should on HT-29 cells metabolism should be in more detail investigated
We agree with the referee’s suggestion; effectively, we are trying to find the necessary grant to perform these type of researches in the next future. Thank you.
References
Ainsworth, E.A., Gillespie, K.M., 2007. Estimation of total phenolic content and other oxidation substrates in plant tissues using Folin-Ciocalteu reagent. Nat. Protoc. 2, 875–877. https://doi.org/10.1038/nprot.2007.102
Berridge, M. V, Herst, P.M., Tan, A.S., 2005. Tetrazolium dyes as tools in cell biology: new insights into their cellular reduction. Biotechnol. Annu. Rev. 11, 127–152. https://doi.org/10.1016/S1387-2656(05)11004-7
Cheng, C., Dong, W., 2018. Aloe-Emodin Induces Endoplasmic Reticulum Stress-Dependent Apoptosis in Colorectal Cancer Cells. Med. Sci. Monit. 24, 6331–6339. https://doi.org/10.12659/MSM.908400
El Sayed, A.M., Ezzat, S.M., El Naggar, M.M., El Hawary, S.S., 2016. In vivo diabetic wound healing effect and HPLC–DAD–ESI–MS/MS profiling of the methanol extracts of eight Aloe species. Brazilian J. Pharmacogn. 26, 352–362. https://doi.org/10.1016/j.bjp.2016.01.009
Gutterman, Y., Chauser-Volfson, E., 2000. The distribution of the phenolic metabolites barbaloin, aloeresin and aloenin as a peripheral defense strategy in the succulent leaf parts of Aloe arborescens. Biochem. Syst. Ecol. 28, 825–838. https://doi.org/10.1016/S0305-1978(99)00129-5
Salehi, B., Albayrak, S., Antolak, H., Kręgiel, D., Pawlikowska, E., Sharifi-Rad, M., Uprety, Y., Tsouh Fokou, P., Yousef, Z., Amiruddin Zakaria, Z., Varoni, E., Sharopov, F., Martins, N., Iriti, M., Sharifi-Rad, J., 2018. Aloe Genus Plants: From Farm to Food Applications and Phytopharmacotherapy. Int. J. Mol. Sci. 19, 2843. https://doi.org/10.3390/ijms19092843
Stockert, J.C., Blázquez-Castro, A., Cañete, M., Horobin, R.W., Villanueva, Á., 2012. MTT assay for cell viability: Intracellular localization of the formazan product is in lipid droplets. Acta Histochem. 114, 785–796. https://doi.org/10.1016/j.acthis.2012.01.006

Reviewer 2 Report
The manuscript describes the antiglycation activity and epithelial cellular uptake of two Aloe arborescens extracts as well as aloin and aloe-emodin. In general, the work is well presented with scientifically sound methods and disussions of the results. The following minor questions/changes should be addressed:
1) Although previous studies have used epithelial cellular uptake as an indication of absorption/bioavailability potential of compounds, it should be taken into consideration that epithelial cellular uptake is not equal to permeation or drug absorption. Lipophilic compounds are especially tending to accumulate in the cells without permeating across the cell monolayer and therefore do not provide good bioavailability. Transport studies across epithelial cell monolayers give a better indication of the absorption potential of compounds and this limitation of the current study should be mentioned somewhere in the paper.
2) Page 9, line 182: The loss of the compounds (aloin and aloe-emodin) during the cellular uptake studies is explained by potential metabolism, but non-specific binding can also contribute significantly to this phenomenon and should be mentioned in addition to the current explanation.
3) Page 3, line 86: Has the stability of the phytochemicals of interest been shown over a period of 5 days in solution?
4) Some minor spelling errors should be corrected as follows: page 3, line 83: A. arborescens should be in italic;, page 3, line 91: cromatograms should be chromatograms and were should be where page 3, ;line 102: compenent should be component.
Author Response
Thank you for your helpful comments and for taking the time to point out suggestions to improve our manuscript.
Although previous studies have used epithelial cellular uptake as an indication of absorption/bioavailability potential of compounds, it should be taken into consideration that epithelial cellular uptake is not equal to permeation or drug absorption. Lipophilic compounds are especially tending to accumulate in the cells without permeating across the cell monolayer and therefore do not provide good bioavailability. Transport studies across epithelial cell monolayers give a better indication of the absorption potential of compounds and this limitation of the current study should be mentioned somewhere in the paper.
- Thank you for your remarks; we agree with the suggestion: our experiments aim firstly to evaluate the possibility that aloin and aloe-emodin could pass through cellular membrane; they are not an absolute demonstration of bioavailability that can be obtained only with clinical studies. Furthermore, we agree that monolayer cell epithelium is a standard approach to study permeability, such as Caco-2 cell monolayer, but in the present detection we chose to use HT-29 cells also to compare the data with the vitality assay. Further, studies with in vitro investigation using isolated rat ileal segments could be performed. In the present research, we used HT-29 cells uptake which is also a reliable and largely used model useful to these purposes. Overall, the mechanisms associated with natural compounds internalization are still highly contentious, and there is little information regarding the epithelial cellular uptake and over their bioavailability. Therefore, in the present study, we conducted an in vitro uptake study using HT-29 cells as an intestinal model to argue preliminary information of in vivo bioavailability. The follow statement was added to discussion (P.10 L.536-537): “However, further in vitro assessments, e.g. using intestinal epithelial cell monolayers, and also in vivo studies are necessary for the evaluation of aloe-emodin and aloin bioavailability.”
Page 9, line 182: The loss of the compounds (aloin and aloe-emodin) during the cellular uptake studies is explained by potential metabolism, but non-specific binding can also contribute significantly to this phenomenon and should be mentioned in addition to the current explanation.
- We agree with the suggestion and a specific sentence was added in the Results section of the manuscript (P.7 L.430-431): “Further, some biases such as compounds sequestration from non-specific binding sites or other types of interactions may have occurred.”
Page 3, line 86: Has the stability of the phytochemicals of interest been shown over a period of 5 days in solution?
The stability of compounds in solution was verified using HPLC analysis; it was observed that aloin and aloe-emodin peaks of chromatograms remain unchanged at least for 8 days. We therefore decided to keep the solutions for a week, precisely, from Monday to Friday. This procedure has also been chosen based on the fact that various commercial preparations containing Aloe are stored for several days before oral administration in therapy.
Some minor spelling errors should be corrected as follows: page 3, line 83: A. arborescens should be in italic;, page 3, line 91: chromatograms should be chromatograms and were should be where page 3, ;line 102: component should be component.
Thank you for pointing out the typographical errors in the manuscript which were corrected with attention.

Reviewer 3 Report
In the current manuscript, Froldi et al report of the biological activity, phytochemicals and safety of Aloe arborescens and its active component. The study was properly designed and the conclusion was based on their findings. However, consider the following comments to improve the manuscript. Additional comments are also included in the annotated PDF.
Suggested references for this work (just a suggestion)
Amoo SO, Aremu AO, Van Staden J (2014) Unraveling the medicinal potential of South African Aloe species. Journal of Ethnopharmacology 153:19-41. doi:http://dx.doi.org/10.1016/j.jep.2014.01.036
Grace OM, Simmonds MSJ, Smith GF, Van Wyk AE (2008) Therapeutic uses of Aloe L. (Asphodelaceae) in southern Africa. Journal of Ethnopharmacology 119:604-614
Kawai K, Beppu H, Shimpo K, Chihara T, Yamamoto N, Nagatsu T, Ueda H, Yamada Y (1998) In vivo effects of Aloe arborescens Miller var. natalensis Berger (Kidachi aloe) on experimental tinea pedis in guinea-pig feet. Phytotherapy Research 12:178-182. doi:10.1002/(sici)1099-1573(199805)12:3<178::aid-ptr218>3.0.co;2-f
Matsuda Y, Yokohira M, Suzuki S, Hosokawa K, Yamakawa K, Zeng Y, Ninomiya F, Saoo K, Kuno T, Imaida K (2008) One-year chronic toxicity study of Aloe arborescens Miller var. natalensis Berger in Wistar Hannover rats. A pilot study. Food and Chemical Toxicology 46:733-739. doi:10.1016/j.fct.2007.09.107
COMMENTS/CONCERNS
1. Title: need amendment to read better and convey the message of the message. For e.g. AntiGlycation Activity and HT-29 Cellular Uptake of Aloe-emodin, Aloin and Aloe arborescens leaf Extracts
2. Ensure all abbreviations are well defined at the first mention (including in the abstract)
3. Line 21: you can only use the word ‘significantly’ based on statistical evidence, if not, then use another appropriate word
4. Fig 2: what value does this add to the manuscript? Can this just be a supplementary materials? It is just an evidence from the analysis and support your findings.
5. Tables 1 and 2 can be easily subjected to statistical analysis (in this case, student’s t-test will do as it compares between 2 treatments/options)
6. Fig 4: You need to calculate the IC50 value for the extracts and compounds. Then you can easily subject the IC50 to appropriate statistical analysis. For instance, use student’s t test when comparing two options. The presentation of value on % had no real value when you have tested across a range of concentrations.
7. The use of additional statistical analysis (Student’s t-test) must be reflected in section 4.8.
8. REFERENCES: Need more attention, all scientific names must be provided in italics
9. Also, ensure all the scientific names are italics throughout the manuscript
10. Consider if you can avoid repeating too much of the data from Tables and Figures in the result section. Only provide the trend. This result section appears wordy as it stands.
Author Response
Thank you very much for all the suggestions; also the references were taken into consideration. Regarding notes on the pdf manuscript, we did not find them, perhaps they were into the file downloaded from the online "Molecules" link.
About suggested references, Amoo et al., 2014 was added in the introduction ( P.1 L.36).
Title: need amendment to read better and convey the message of the message. For e.g. AntiGlycation Activity and HT-29 Cellular Uptake of Aloe-emodin, Aloin and Aloe arborescens leaf Extracts.
Effectively the title proposed fits very well with our aims and therefore our title was changed as suggested. Thank you very much for the valuable suggestion.
2. Ensure all abbreviations are well defined at the first mention (including in the abstract)
All abbreviations have been defined either in the abstract and when firstly appeared in the manuscript.
3. Line 21: you can only use the word ‘significantly’ based on statistical evidence, if not, then use another appropriate word
The word ‘significantly’ was changed with “substantially”, (P.1 L.22).
4. Fig 2: what value does this add to the manuscript? Can this just be a supplementary materials? It is just an evidence from the analysis and support your findings.
We agree with the suggestion, figure 2 was moved to supplementary materials (Figure S2) and the sentences reporting aloin and aloe-emodin retention times were inserted in the materials section (P.10 L.576-580).
5. Tables 1 and 2 can be easily subjected to statistical analysis (in this case, student’s t-test will do as it compares between 2 treatments/options)
The statistical analysis of the experimental data was improved (Tables 1 and 2; and figures) and Student’s t-test was performed where appropriate, (P.5 L.314; 317-318 also Figure 3). Thank you for the advice.
6. Fig 4: You need to calculate the IC50 value for the extracts and compounds. Then you can easily subject the IC50 to appropriate statistical analysis. For instance, use student’s t test when comparing two options. The presentation of value on % had no real value when you have tested across a range of concentrations.
We agree with the suggestion and where it was possible the EC50 (or ever EC20) were estimated and reported in the text (P.5 L.314; 317-318 and also Figure 3). However, we still used the percentage of maximum effect because the extracts and the compounds did not induce the maximum of the effect (100%) and there was no other way to compare them.
7. The use of additional statistical analysis (Student’s t-test) must be reflected in section 4.8.
- The section 4.8 was changed as suggested. The effects data were analysed using GraphPad Prism 5, and the EC50 values were calculated (P.12 L.640-643).
REFERENCES: Need more attention, all scientific names must be provided in italics
Thank you. The changes were done; all scientific names are in italics.
9. Also, ensure all the scientific names are italics throughout the manuscript
Thank you. We checked all the scientific names throughout the manuscript changing them in italics.
10. Consider if you can avoid repeating too much of the data from Tables and Figures in the result section. Only provide the trend. This result section appears wordy as it stands.
We tried to change the results section improving the text. Thank you.

Round 2
Reviewer 1 Report
Although the authors have tried to answer the various questions I have addressed in the first review, they did not take into account my recommendations to improve the manuscript.
Here some results are not convincing enough and must be validated by alternative methods. In addition, no clarification with regard to my questions has been added to the text.
In particular, with regard to the method used to assess cellular toxicity: the authors explain that MTT is the most commonly used method to assess cytotoxicity. Certainly, but in most valid studies, mortality is also evaluated by another method.
Similarly, the authors found a convincing antiglycant effect and dose-dependent by the AGE fluorescence method but this is not sufficient to conclude. Several alternative methods not time and money consuming could be used such as fructosamine assay, DTNB or TNBS assays (Diabetes Metab. 2012 38(2):171-8) could be used.
In conclusion no real improvement have been made.
Author Response
We are very regretted for the unfavourable consideration on our manuscript which we tried to improve in the revised format.
The reviewer asked us new experiments, with different protocols which surely may be improve the research; but, unfortunately, this research is concluded in the laboratory and we must work on other topics. Further, we cannot performe further researches on this matter because our Department changed the goals. We are strongly sure about our experimental protocols even if, we agree, new sets of experiments may improve the present research. We thank the reviewer for the suggestions.
We hope that the reviewer could overall consider positively our research, supporting the manuscript for publication in Molecules, also because we believe that the present study could be of interest to other researchers in various fields.
Anyway, thank you for your remarks that we will take in mind for future research plans.
Thank you very much for your attention.
GF

Round 3
Reviewer 1 Report
In my both reviews, I asked for additional experiments to be carried out. For me, some conclusions cannot be supported by a single technique. The authors argue that the study is completed to justify the refusal to follow the referee's recommendations.
In my opinion, a study is really finished when the results are published and generally speaking when submitting work for publication, we should expect to have to do additional work.
With regard to this study, I have already made my recommendations. I leave it to the authors not to follow my recommendations and therefore to the publisher to make the appropriate decision